# Facile Synthesis of Methylsilsesquioxane Aerogels with Uniform Mesopores by Microwave Drying

**DOI:** 10.3390/polym11020375

**Published:** 2019-02-20

**Authors:** Xingzhong Guo, Jiaqi Shan, Wei Lei, Ronghua Ding, Yun Zhang, Hui Yang

**Affiliations:** 1School of Materials Science and Engineering, Zhejiang University, Hangzhou 310027, China; 21626008@zju.edu.cn (J.S.); yanghui@zju.edu.cn (H.Y.); 2Pan Asia Microvent Tech (Jiangsu) Coporation & Zhejiang University Micro-nano-porous Materials Joint Research Development Center, Changzhou 213100, China; leiwei@microwent.com.cn (W.L.); dingronghua@microvent.com.cn (R.D.); zhangyun@microvent.com.cn (Y.Z.)

**Keywords:** MSQ aerogel, mesoporous structure, sol–gel, microwave drying

## Abstract

Methylsilsesquioxane (MSQ) aerogels with uniform mesopores were facilely prepared via a sol–gel process followed by microwave drying with methyltrimethoxysilane (MTMS) as a precursor, hydrochloric acid (HCl) as a catalyst, water and methanol as solvents, hexadecyltrimethylammonium chloride (CTAC) as a surfactant and template, and propylene oxide (PO) as a gelation agent. The microstructure, chemical composition, and pore structures of the resultant MSQ aerogels were investigated in detail to achieve controllable preparation of MSQ aerogels, and the thermal stability of MSQ aerogels was also analyzed. The gelation agent, catalyst, solvent, and microwave power have important roles related to the pore structures of MSQ aerogels. Meanwhile, the microwave drying method was found to not only have a remarkable effect on improving production efficiency, but also to be conducive to avoiding the collapse of pore structure (especially micropores) during drying. The resulting MSQ aerogel microwave-dried at 500 W possessed a specific surface area up to 821 m^2^/g and a mesopore size of 20 nm, and displayed good thermal stability.

## 1. Introduction

Silica aerogel is a low-density porous solid material with three-dimensional reticulate skeletons formed by interconnected SiO_2_ nanoparticles [1]. Benefiting from its special structure, silica aerogel possesses various unique properties [2,3,4,5] such as high specific surface area and porosity, low thermal conductivity, high visible-light transmittance, and low dielectric constant. Owing to these properties, silica aerogel was applied in various fields such as catalyst carriers [6,7,8], heat insulation [9,10,11], acoustic insulation [12,13,14], Cherenkov radiation detectors [15], adsorbents [16,17,18,19], and thermal energy storage [20]. Silica aerogel was first prepared by Kistler in the 1930 s via supercritical drying [21]. Until now, supercritical drying is still an effective method to avoid aerogel shrinkage and cracks during drying. However, supercritical drying, which needs special conditions such as high pressure and high temperature, prevents aerogels from industrial manufacturing and extended applications. Therefore, the preparation of silica aerogel via ambient pressure drying became a hot research topic in these years. One practical way is to use organoalkoxysilanes with small organic substituent groups to obtain organic–inorganic hybrid aerogel. Methylsilsesquioxane (MSQ) aerogel [22,23,24] is a methyl hybrid silica aerogel, derived from organoalkoxysilanes with methyl groups through a facile sol–gel process [25,26]. The methyl group was introduced in order to enhance the mechanical strength of aerogel skeletons formed by interconnected nanoparticles, which was proven to be effective.

Therefore, in the last few years, lots of research efforts were devoted to the synthesis of silica aerogels by ambient pressure drying. He et al. [27] reported the experimental results on the synthesis of water/glass-based silica aerogels dried under ambient pressure, using *N*,*N*-dimethylformamide (DMF) as a drying control chemical additive (DCCA), trimethylchlorosilane (TMCS) for surface modification, and ethanol and *n*-hexane for solvent replacement, in which DCCA and TMCS used to reduce the capillary during drying were environmentally unfriendly and poisonous, preventing aerogels from extended applications. To avoid using DCCA as a surface modifier, Hayase et al. [28] reported the preparation of transparent MSQ aerogels via a facile sol–gel process, using methanol, 2-propanol, and heptane for multistage solvent replacement, in which a kind of “aging sol” was used to enhance mechanical strength of the aerogel skeleton for reducing the capillary during drying. This preparation method was environmentally friendly but not efficient, as it took an extra three days for preparation of aging sol, soaking of aging sol, and evaporation at room temperature, preventing MSQ aerogels from industrial manufacturing. As well known, microwave drying is extensively used to dry foods, ceramics, chemical products, woods, and other materials due to its high drying rate and uniform drying. Up to now, there are some works reporting the applicaton of microwave drying in silica aerogels. Duraes introduced the preparation of silica-based aerogel-like materials by quick microwave drying, and aerogel-like materials with high hydrophobicity (contact angle ~ 150°) and surface area (414 m^2^/g) can be obtained [29]. Nocentini studied the hygro-thermal properties of silica aerogel blankets dried using microwave heating, and the aerogel blankets can be used in building thermal insulation [30].

In this study, methylsilsesquioxane (MSQ) aerogels were easily prepared via a sol–gel process using methyltrimethoxysilane (MTMS) as a precursor, hydrochloric acid (HCl) as a catalyst, water and methanol as solvents, hexadecyltrimethylammonium chloride (CTAC) as a surfactant and template, and propylene oxide (PO) as a gelation agent, while methanol, 2-propanol, and heptane were used for multistage solvent replacement, followed by microwave drying. The pH, solvent polarity, volume, and microwave power have important roles related to the pore structures of MSQ aerogels, and the controllable preparation of MSQ aerogels with uniform mesopores can be facilely achieved by microwave drying. Meanwhile, we found that microwave drying not only has a remarkable effect on improving production efficiency, but also is conducive to avoiding collapse of the pore structure (especially micropores), resulting from thermal gradients during drying. 

## 2. Materials and Methods

### 2.1. Materials

Methyltrimethoxysilane (MTMS, Aladdin, Shanghai, China, 98%), hydrochloric acid (HCl, Aladdin, Shanghai, China), hexadecyltrimethylammonium chloride (CTAC, Aladdin, Shanghai, China, 97%), methanol (Sinopharm Chemical Reagent Co., Ltd., Shanghai, China, ≥99.5%), propylene oxide (PO, Sinopharm Chemical Reagent Co., Ltd., Shanghai, China, ≥99.5%), methanol (Sinopharm Chemical Reagent Co., Ltd., Shanghai, China, ≥99.7%), 2-propanol (Sinopharm Chemical Reagent Co., Ltd., Shanghai, China, ≥99.7%), *n*-heptane (Sinopharm Chemical Reagent Co., Ltd., Shanghai, China, ≥98.5%) were used as obtained.

### 2.2. Preparation of MSQ Aerogel

Table 1 shows the starting compositons and drying mehods of all MSQ aerogel samples mentionded below in this report. Figure 1 shows the reaction scheme and preparation process of MSQ aerogels. Firstly, 0.24 g of CTAC, methanol, and HCl solution were mixed in a glass tube, and then 3.0 mL of MTMS was added with vigorous stirring in an ice-bath for hydrolysis and polymerization of MTMS. After continuously stirring for 30 min, PO was added to the transparent solution (MSQ sol). After stirring for 1–2 min, the resultant solution was allowed to gelate at 40 °C under closed conditions. The gelation time was about 50 min. The resultant gel was aged over 30 min at the same temperature and firstly solvent exchanged in methanol twice to remove water and CTAC. After methanol solvent replacement, the resultant gel was exchanged in 2-propanol twice and *n*-heptane twice at 60 °C. Then, the wet gel with *n*-heptane was dried by microwave to obtain the MSQ aerogel.

### 2.3. Characterization

The microstructure of MSQ aerogels was observed using a scanning electron microscope (SEM: Su8010, Hitachi, Tokyo, Japan) and transmission electron microscope (TEM: JEM-2100, JEOL, Tokyo, Japan). The chemical compositions were confirmed by Fourier-transform infrared spectroscopy (FT-IR, Nicolet 6700, ThermoFisher Scientific, Waltham, MA, USA), differential thermal analysis (DTA, Q200, TA, New Castle, DE, USA), and thermogravimetry (TG, TA-Q500, TA, New Castle, DE, USA). Pore structures of MSQ aerogels were characterized using an N_2_ adsorption–desorption apparatus (BET, ASAP2020HD88, Micromeritics Instruments Corporation, Norcross, GA, USA), and the samples were degassed at 120 °C under vacuum before each N_2_ adsorption–desorption measurement. The BJH (Barret-Joyner-Halenda) method was applied to the adsorption branch to derive mesopore size distributions.

## 3. Results and Discussion

### 3.1. Controllable Preparation of MSQ Aerogels with Uniform Mesopores

#### 3.1.1. Effect of Gelation Agent (PO) on Pore Structures of MSQ Aerogels

Figure 2a–e show SEM images of MSQ aerogels prepared via varied sol–gel processes with different volumes of gelation agent (PO), and Figure 2f shows the gelation time (*t*) of MSQ aerogels prepared with different volumes of PO. With the increase in PO volume, the gelation time of aerogels decreased until the volume of PO was 1.0 mL, which indicates that 1.0 mL of PO is enough to participate in the ring-opening reaction with HCl for slowly elevating the pH of sol [31]. However, when the volume of PO increased from 1.0 to 1.5 mL, the mesopores of aerogels become larger, due to the extra part of PO indirectly increasing the volume of the liquid phase. Similarly, this phenomenon can be used to explain the change of microstructure of aerogels from Figure 2a to 2b. However, with the increase in PO volume, the gelation time of aerogels had a marked decline, which led to inadequate separation of solid and liquid phases. As a result, we conclude that 1.0 mL of PO is appropriate in the sol–gel preparation of MSQ aerogels.

#### 3.1.2. Effect of Catalyst (HCl) on Pore Structures of MSQ Aerogels

Figure 3a–e show SEM images of MSQ aerogels prepared via varied sol–gel processes with different concentrations of HCl solutions, and Figure 3f shows variation of the velocity of hydrolysis and polymerization with pH [32,33]. According to the mechanism of hydrolysis and polymerization reactions of MTMS, the smaller the difference between velocity of hydrolysis and polymerization (∆*V*) is, the smaller the nanoparticles of aerogel will be. In other words, there must be a point of pH with which the nanoparticles of aerogel prepared are smallest. However, when the pH value increased from 1.3 to 3.3 in this sol–gel system, the microstructure of MSQ aerogel changed little, due to the hydrolysis and polymerization reactions of MTMS being carried out under ice-bath conditions in this process. Both the velocity of hydrolysis and polymerization resulted in little change of ∆*V*. Figure 4 shows N_2_ adsorption–desorption isotherms (a) and BJH mesopore size distributions (b) of MSQ aerogels prepared with different pH values, and the pore structures of MSQ aerogels are listed in Table 2. As observed from Figure 4 and Table 2, with the increase of pH, the isotherms, mesopore size distributions, and pore structure data of aerogels also changed little, which corresponds to Figure 3a–e. In this sol–gel system, the reaction conditions in the ice-bath not only avoided the temperature effect caused by the exothermic hydrolysis reaction, but also reduced the effect of catalyst concentration on the structure of MSQ aerogels, which is advantageous to improving the stability of MSQ aerogel production. 

#### 3.1.3. Effect of Solvent on Pore Structures of MSQ Aerogels

Figure 5a–f show SEM images of MSQ aerogels prepared by 350-W microwave drying via varied sol–gel processes with different solvent polarity. Upon comparing Figure 5a with 5f, it was found that, without the addition of methanol, the microstructure of MSQ aerogel became more compact. In this sol–gel process, methanol is an indispensable solvent not only to dissolve and disperse MTMS and CTAC, but also to restrain the hydrolysis and condensation of MTMS. With the increase in volume ratios of methanol and water (*M*/*W*), the sizes of mesopores and nanoparticles of resultant MSQ aerogels successively became smaller because of successively decreasing phase separation in their sol–gel processes. According to Flory–Huggins lattice theory [36], in a two-phase mixed system, the larger the polarity difference is between the two phases, the larger the phase separation trend will be. Thus, with the increase of *M*/*W*, the polarity difference between component solvent and sol derived from MTMS became small, indicating a low phase separation. Therefore, it can be concluded that it is effective to achieve controllable preparation of MSQ aerogels by adjusting polarity of the component solvent. Figure 6a–e show TEM images of MSQ aerogels prepared with different solvent polarity, and Figure 6f shows a diagram of phase separation [37]. The changes in Figure 6a–e are in accordance with the process of phase separation in Figure 6f, which further demonstrates that the solvent polarity determines the degree of phase separation in this sol–gel system, which is reliable for controllable preparation of MSQ aerogels. Figure 7 shows N_2_ adsorption–desorption isotherms (a) and BJH mesopore size distributions (b) of MSQ aerogels prepared by 350-W microwave drying via varied sol–gel processes with different solvent polarity. As the *M*/*W* increased from 0.33 to 3.0, the values of volume absorbed and pore diameter gradually decreased, resulting from an increasing trend of phase separation between component solvent and skeletons, which is consistent with the microstructure characterization above (Figure 5). Table 3 shows pore structures of MSQ aerogels prepared by 350-W microwave drying via varied sol–gel processes with different solvent polarity. The specific surface area of MSQ aerogel without methanol was far less than that of MSQ aerogels with methanol, which is also consistent with the microstructure characterization above. 

Figure 8a–e show SEM images of MSQ aerogels prepared by 350-W microwave drying via varied sol–gel processes with different solvent volumes. As is known, the pores of aerogels result from the evaporation of solvent. Logically, with the increase in solvent volume, the pore size of MSQ aerogels became larger, corresponding to Figure 8b–e. However, the microstructure of MSQ was too compact, whereby the specific surface area and micropore surface area were much lower than other samples observed from Table 4. This is because, in this sol–gel process, 1 mL of solvent was not enough to adequately dissolve and disperse 3 mL of MTMS and 0.24 g of CTAC, leading to the agglomeration of sol particles derived from MTMS and micelles derived from CTAC. By adjusting volume of solvent to control the pore size of MSQ aerogels, we can achieve the controllable preparation of MSQ aerogels. Figure 9 shows N_2_ adsorption–desorption isotherms (a) and BJH mesopore size distributions (b) of MSQ aerogels prepared by 350-W microwave drying via varied sol–gel processes with different solvent volumes. The isotherm and BJH mesopore size distribution of MSQ aerogel with 1 mL of solvent further confirms its impact on microstructure, as characterized by SEM above. As the solvent volume increased, the pore diameter of MSQ aerogels successively increased, and the pore structure of MSQ aerogels evolved from uniform mesopores to macropores, which is consistent with the result of SEM images in Figure 8. 

#### 3.1.4. Effect of Microwave Power on Pore Structures of MSQ Aerogels

To optimize the process of microwave drying and study the influence of microwave power on microstructure of MSQ aerogel, MSQ aerogels prepared by microwave drying at different powers, oven drying, and indoor evaporation were characterized by SEM and the N_2_ adsorption–desorption method. Figure 10a–d show SEM images of MSQ aerogels prepared by microwave drying at different powers, and Figure 10e,f show SEM images of MSQ aerogel prepared by 40 °C oven drying and indoor evaporation, respectively. It was observed that the MSQ aerogels prepared by microwave drying had more uniform mesopores than MSQ aerogels prepared by oven drying and evaporation, which benefited from the entirety of microwave drying. Also, it can be concluded that the power of the microwave had little influence on the microstructure of resultant MSQ aerogels in this experiment drying only 10 cm^3^ of wet gel. Figure 11a shows N_2_ adsorption–desorption isotherms of MSQ aerogels prepared by varied drying methods. According to the classification of international union of pure and applied chemistry (IUPAC), all isotherms of MSQ aerogels prepared by varied drying methods belonged to type IV with hysteresis loops of type H2, which proves that all the resultant MSQ aerogels had uniform ampuliform mesopores. This shape of mesopores resulted from skeletons of MSQ aerogel formed by point-connected spherical nanoparticles. Figure 11b shows BJH mesopore size distributions of MSQ aerogels prepared by varied drying methods. It was found that the mesopores of MSQ aerogels prepared by 700-, 500-, and 350-W microwave drying mainly distributed at 20 nm, while the mesopores of MSQ aerogels prepared by 200-W microwave drying, oven drying, and indoor evaporation mainly distributed at 30 nm. As is known, the higher the microwave power is, the faster the rate of microwave heating will be. Therefore, it is believed that too fast a heating rate during microwave drying leads to the irreversible shrinkage of mesopores from 30 to 20 nm. Table 5 shows the pore structures of MSQ aerogels prepared by varied drying methods. The specific surface areas of MSQ aerogels prepared by microwave drying were as high as those prepared by oven drying and evaporation, while the time of microwave drying (less than 1 h) was far less than the time of oven drying (24 h) and evaporation (more than 48 h). In particular, MSQ aerogel prepared by 500-W microwave drying possessed a specific surface area up to 821 m^2^/g and a micropore surface up to 617 m^2^/g, which proves that microwave drying was more effective in reducing thermal gradients during the drying process than oven drying, where the resultant micropore surface was only 491 m^2^/g.

### 3.2. Thermal Stability of Typical MSQ Aerogel

Figure 12a shows differential thermal analysis (DTA) and thermogravimetry (TG) curves of a typical MSQ aerogel prepared by microwave drying (sample S20). When the temperature rose to 200 °C, a mass loss of 3% occurred on the TG curve with no peak on the DTA curve owing to the slow evaporation of absorbed water. When the temperature rose to 400 °C, methyl groups of MSQ aerogel were oxygenated, which resulted in an exothermic peak on the DTA curve and a mass loss of 6% on the TG curve. Without the exothermic peak of other reagents, DTA/TG curves prove the high purity of the resultant MSQ aerogel. Figure 12b shows the infrared (IR) spectrum of the as-prepared MSQ aerogel. The absorption peaks near 3350, 3470, and 3410 cm^−1^ were assigned as O–H stretches of absorbed water, and a small absorption peak near 1630 cm^−1^ was attributed to the bending vibration of the O–H group, which agrees with the results of DTA/TG curves. Small absorption peaks near 2970 cm^−1^ were the result of asymmetric and symmetric stretching vibrations of the C–H group, and two peaks near 1275 and 780 cm^−1^ corresponded to the symmetric bending and rocking vibration of the Si–CH_3_ group. The absorption peaks near 1140, 1040, and 445 cm^−1^ were attributed to O–Si–O bending. These results indicate the formation of methylsilsesquioxane with a methyl group. 

As is known, thermal stability is always an important indicator for aerogels, because aerogels are usually applied in certain conditions of temperature, such as for applications of heat insulation, catalyst carriers, thermal energy storage, and so on. Therefore, heat treatment at 200–800 °C of the typical MSQ aerogel prepared by 500-W microwave drying (sample S20) was conducted to investigate the thermal stability of the mesoporous structure, as well as the influence of high temperature on the surface area and micropores of the MSQ aerogel. Figure 13 shows SEM images of MSQ aerogels at varied heat treatment temperatures. With the increase of heat treatment temperature, the pore structure of MSQ aerogel gradually decreased, and it became more compact, especially at 400, 600, and 800 °C. As observed from the SEM images of the aerogel, the collapse of the holistic pore structure occurred between 300 and 400 °C, which corresponds to the temperature of methyl decomposition inferred above in Figure 12a. This suggests that it is the exothermic reaction of methyl decomposition which resulted in the collapse of the holistic pore structure. Figure 14 shows the N_2_ adsorption–desorption isotherms (a) and BJH mesopore size distributions (b) of MSQ aerogels at varied heat treatment temperatures, and Table 6 shows the pore structure data of the resultant MSQ aerogels. At 200 and 300 °C, the volume absorbed at high relative pressure of aerogel decreased, and the volume absorbed at low relative pressure changed little corresponding to little decrease in the specific surface area of the aerogel and micropore surface, which resulted from the shrinkage of mesopores. When heat temperature increased to 400 °C, both volume absorbed at low and high relative pressure and specific surface area of the aerogel decreased a lot, indicating the collapse of the holistic pore structure due to the exothermic reaction of methyl decomposition, which corresponds to the SEM images in Figure 13. When the heat treatment was 800 °C, the typical MSQ aerogel prepared by 500-W microwave drying still had a specific area of 322 m^2^/g and a micropore surface of 210 m^2^/g.

### 3.3. Discussion on Microwave Drying Mechanism of MSQ Aerogel

Microwave drying is a new drying method for aerogels, and there are still lots of factors to be discussed and researched due to its complicated mechanism. Figure 15 shows the mechanism diagram of microwave drying and oven drying. Different from oven drying, the heat source of microwave drying is evenly distributed within the sample, due to the microwave heating being achieved by an increase in the overall molecular vibration, resulting from the interaction between electric dipoles of the solvent and the alternating electromagnetic field of the microwave. Therefore, the temperature of the different parts of the sample is equal during drying. On the other hand, the heat source of oven drying is the air in the oven; thus, there is a heat conduction process from air to the inside of sample, which results in a temperature gradient in the sample. This temperature gradient is one of the main reasons for the collapse and shrinkage of pore structure during oven drying. Because the heat source of microwave drying is the sample itself, there is still a heat conduction process from sample to air. In our point of view, although this heat conduction process has little influence on the drying effect, it may have a serious impact on efficiency. It is easy to comprehend that, when the temperature of air is lower and the size of the microwave drying chamber is a lot larger than the sample, the frequency of microwave drying should be higher to achieve the drying of MSQ aerogel in the same time. Thus, in order to achieve the quick drying of MSQ aerogel by microwave drying, size matching between the microwave drying chamber and the sample is essential.

## 4. Conclusions

Methylsilsesquioxane aerogels with uniform mesopores were facilely prepared via a sol–gel process by microwave drying. The influences of gelation agent, catalyst, solvent, and microwave power on pore structure were investigated in detail to achieve controllable preparation of MSQ aerogels, and the thermal stability of MSQ aerogels was also analyzed. When the molar ratio of MTMS:HCl:H_2_O:MeOH:CTAC:PO was 1:7.13 × 10^−4^:3.96:1.76:0.035:0.68, and the microwave power was 500 W, MSQ aerogel with a specific surface area of 821 m^2^/g and a mesopore size of 20 nm can be obtained. After heat treatment of 800 °C, the microwave-dried MSQ aerogel still had a specific area of 322 m^2^/g and a micropore surface of 210 m^2^/g. To pave the way for the follow-up researches on microwave drying, the mechanism of microwave drying was discussed in detail, and size matching between the microwave drying chamber and the sample is considered essential for drying efficiency.

## Figures and Tables

**Figure 1 polymers-11-00375-f001:**
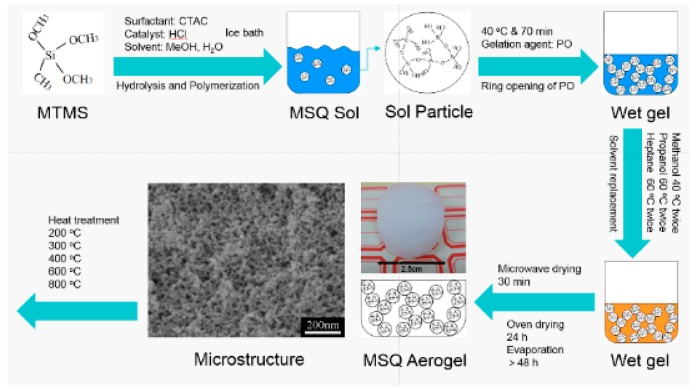
Reaction scheme and preparation process of methylsilsesquioxane (MSQ) aerogels.

**Figure 2 polymers-11-00375-f002:**
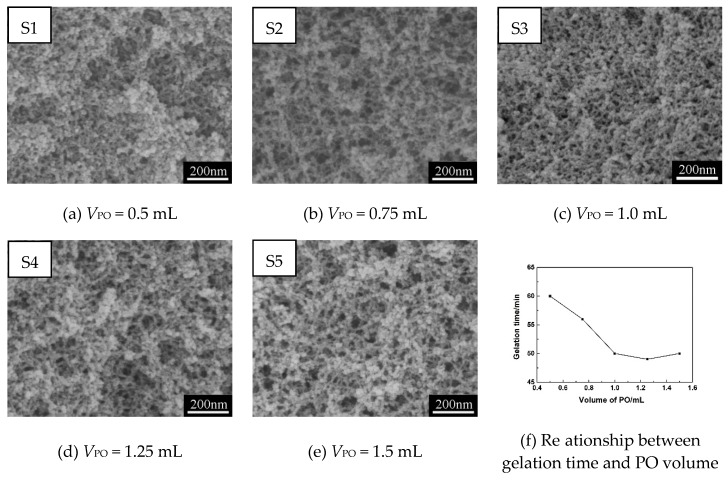
SEM images (**a**–**e**) of MSQ aerogels prepared by 350-W microwave drying via varied sol–gel processes with different propylene oxide (PO) volumes (*V*_PO_), and (**f**) gelation time (*t*) of MSQ aerogels prepared by 350-W microwave drying via varied sol–gel processes with different solvent volumes (*V*_PO_).

**Figure 3 polymers-11-00375-f003:**
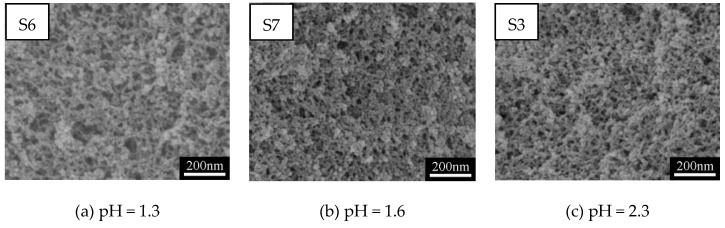
SEM images (**a**–**e**) of MSQ aerogels prepared by 350-W microwave drying via varied sol–gel processes with different pH conditions, and (**f**) hydrolysis and polymerization mechanism of MSQ aerogel [32,33].

**Figure 4 polymers-11-00375-f004:**
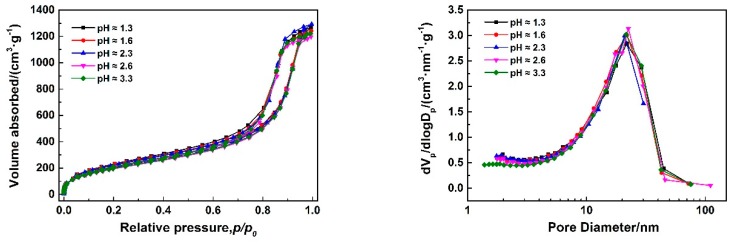
N_2_ adsorption–desorption isotherms (**a**) and BJH mesopore size distributions (**b**) of MSQ aerogels prepared by 350-W microwave drying via varied sol–gel processes with different pH values.

**Figure 5 polymers-11-00375-f005:**
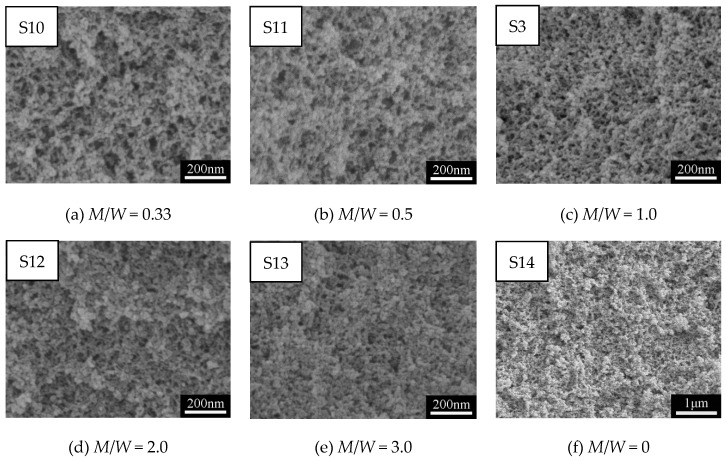
SEM images of MSQ aerogels prepared by 350-W microwave drying via varied sol–gel processes with different solvent polarity. Solvent polarity was adjusted by changing volume ratios of methanol (*M*) and water (*W*) (*M*/*W*).

**Figure 6 polymers-11-00375-f006:**
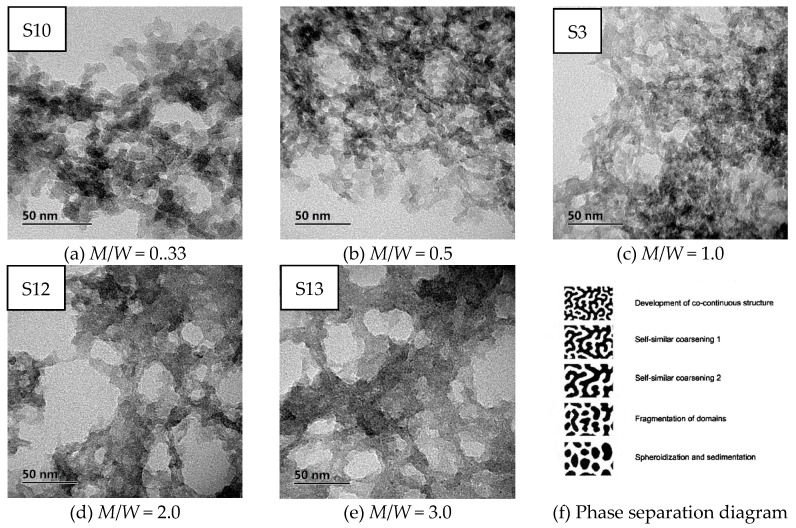
TEM images (**a**–**e**) of MSQ aerogels prepared by 350-W microwave drying via varied sol–gel processes with different solvent polarity. Solvent polarity was adjusted by changing volume ratios of methanol and water (*M*/*W*). A diagram of phase separation is shown in (**f**) [37].

**Figure 7 polymers-11-00375-f007:**
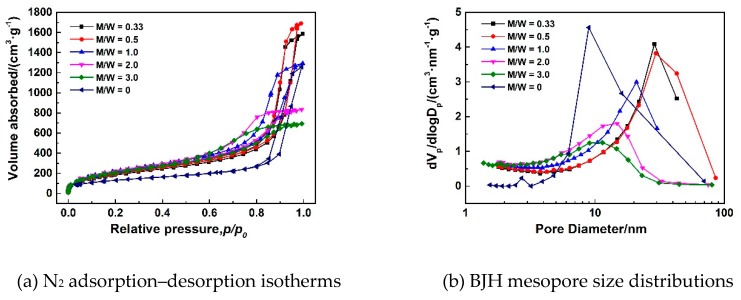
N_2_ adsorption–desorption isotherms (**a**) and BJH mesopore size distributions (**b**) of MSQ aerogels prepared by 350-W microwave drying via varied sol–gel processes with different solvent polarity. Solvent polarity was adjusted by changing volume ratios of methanol and water (*M*/*W*).

**Figure 8 polymers-11-00375-f008:**
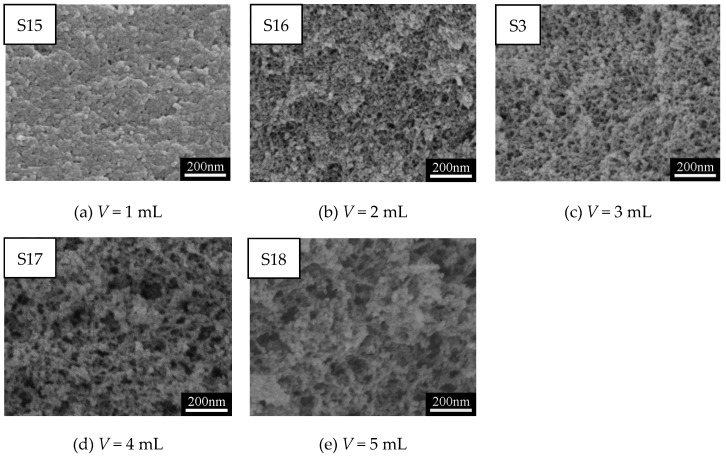
SEM images of MSQ aerogels prepared by 350-W microwave drying via varied sol–gel processes with different solvent volume (V).

**Figure 9 polymers-11-00375-f009:**
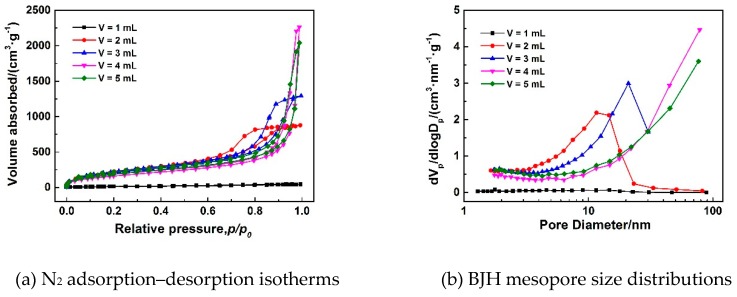
N_2_ adsorption–desorption isotherms (**a**) and BJH mesopore size distributions (**b**) of MSQ aerogels prepared by 350-W microwave drying via varied sol–gel processes with different solvent volume (*V*).

**Figure 10 polymers-11-00375-f010:**
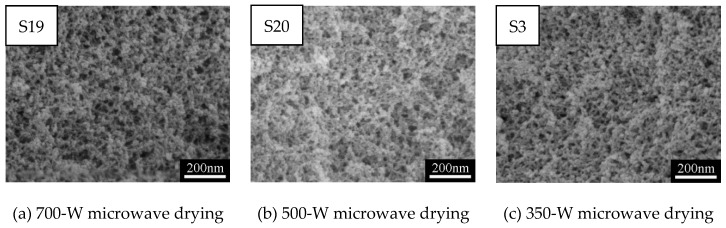
SEM images of MSQ aerogels prepared by varied microwave powers.

**Figure 11 polymers-11-00375-f011:**
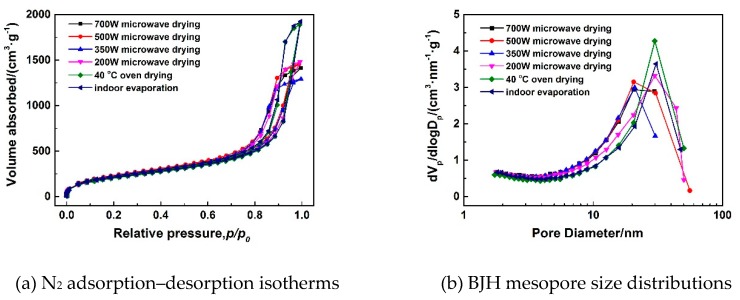
N_2_ adsorption–desorption isotherms (**a**) and BJH mesopore size distributions (**b**) of MSQ aerogels prepared by varied drying methods.

**Figure 12 polymers-11-00375-f012:**
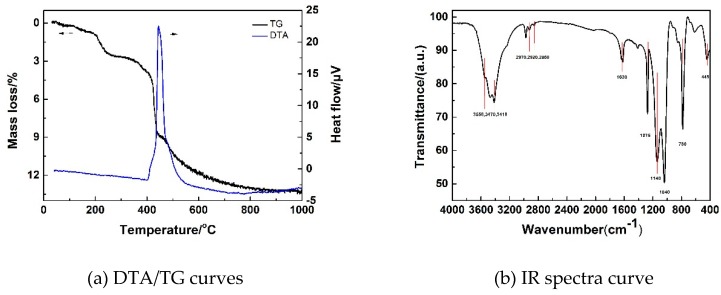
Differential thermal analysis (DTA)/thermogravimetry (TG) curves (**a**) and infrared (IR) spectrum (**b**) of a typical MSQ aerogel prepared by microwave drying.

**Figure 13 polymers-11-00375-f013:**
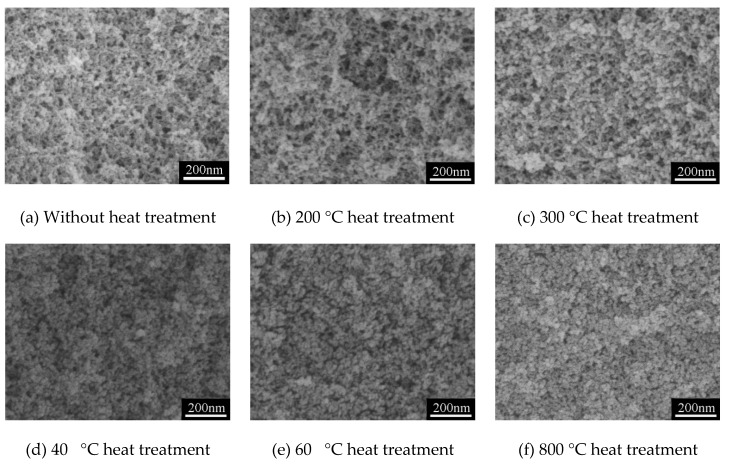
SEM images of MSQ aerogels at varied heat treatment temperatures.

**Figure 14 polymers-11-00375-f014:**
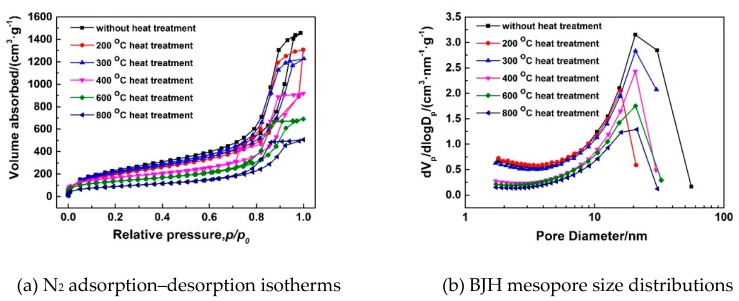
N_2_ adsorption–desorption isotherms (**a**) and BJH mesopore size distributions (**b**) of MSQ aerogels at varied heat treatment temperature.

**Figure 15 polymers-11-00375-f015:**
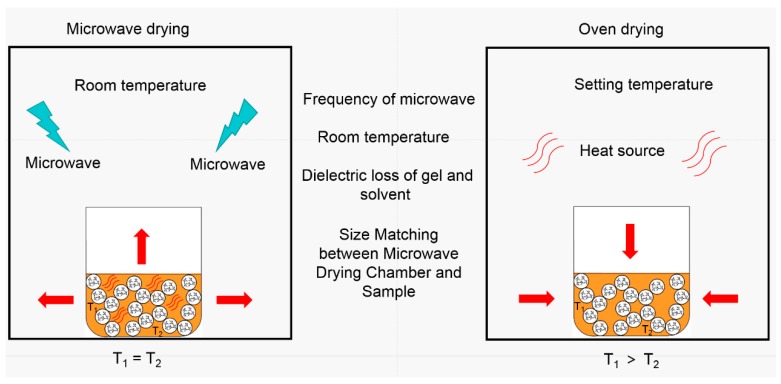
Diagram of microwave drying and oven drying.

**Table 1 polymers-11-00375-t001:** The starting compositons and drying mehods of all methylsilsesquioxane (MSQ) aerogel samples.

Sample	V_MTMS_ (mL)	V_HCl_ (mL)	C_HCl_ (mol/L)	V_MeOH_ (mL)	M_CTAC_ (g)	V_PO_ (mL)	Drying Method
1	3	1.5	0.01	1.5	0.24	0.5	350 W microwave
2	3	1.5	0.01	1.5	0.24	0.75	350 W microwave
3	3	1.5	0.01	1.5	0.24	1.0	350 W microwave
4	3	1.5	0.01	1.5	0.24	1.25	350 W microwave
5	3	1.5	0.01	1.5	0.24	1.5	350 W microwave
6	3	1.5	0.1	1.5	0.24	1.0	350 W microwave
7	3	1.5	0.05	1.5	0.24	1.0	350 W microwave
8	3	1.5	0.005	1.5	0.24	1.0	350 W microwave
9	3	1.5	0.001	1.5	0.24	1.0	350 W microwave
10	3	2.25	0.00667	0.75	0.24	1.0	350 W microwave
11	3	2.0	0.0075	1.0	0.24	1.0	350 W microwave
12	3	1.0	0.015	2.0	0.24	1.0	350 W microwave
13	3	0.75	0.02	2.25	0.24	1.0	350 W microwave
14	3	3	0.01	0	0.24	1.0	350 W microwave
15	3	0.5	0.01	0.5	0.24	1.0	350 W microwave
16	3	1.0	0.01	1.0	0.24	1.0	350 W microwave
17	3	2.0	0.01	2.0	0.24	1.0	350 W microwave
18	3	2.5	0.01	2.5	0.24	1.0	350 W microwave
19	3	1.5	0.01	1.5	0.24	1.0	700 W microwave
20	3	1.5	0.01	1.5	0.24	1.0	500 W microwave
21	3	1.5	0.01	1.5	0.24	1.0	250 W microwave
22	3	1.5	0.01	1.5	0.24	1.0	40 °C oven
23	3	1.5	0.01	1.5	0.24	1.0	evaporation

*C*_HCl_: concentration of HCl solution; *V*_MTMS_: volume of MTMS; *V*_HCl_: volume of HCl solution; *V*_MeOH_: volume of methanol; *V*_PO_: volume of PO; *M*_CTAC_: mass of CTAC.

**Table 2 polymers-11-00375-t002:** Pore structure data of MSQ aerogels prepared by 350-W microwave drying via varied sol–gel processes with different pH values.

pH	S_p_ ^a^ (m^2^/g)	V_pore_ ^b^ (cc/g)	S_micro_ ^c^ (m^2^/g)	V_micro_ ^c^ (cc/g)
1.3	786	1.9	588	0.3
1.6	789	1.9	580	0.3
2.3	784	1.9	596	0.3
2.6	731	1.8	547	0.3
3.3	759	1.8	552	0.3

^a^ Brunauer–Emmett–Teller specific surface area. ^b^ Pore volume calculated by non-local density function theory (NLDFT) method [34] from the adsorption branch. ^c^ Micropore surface and volume calculated by the Dubinin–Astakhov method [35].

**Table 3 polymers-11-00375-t003:** Pore structure data of MSQ aerogels prepared by 350-W microwave drying via varied sol–gel processes with different solvent polarity. Solvent polarity was adjusted by changing volume ratios of methanol and water (*M*/*W*).

M/W	S_p_ ^a^ (m^2^/g)	V_pore_ ^b^ (cc/g)	S_micro_ ^c^ (m^2^/g)	V_micro_ ^c^ (cc/g)
0.33	702	2.4	494	0.2
0.5	741	2.6	538	0.2
1.0	783	1.9	596	0.3
2.0	782	1.2	575	0.3
3.0	772	1.0	567	0.3
0	472	1.9	52	0.04

^a^ Brunauer–Emmett–Teller specific surface area. ^b^ Pore volume calculated by NLDFT method [34] from the adsorption branch. ^c^ Micropore surface and volume calculated by the Dubinin–Astakhov method [35].

**Table 4 polymers-11-00375-t004:** Pore structure data of MSQ aerogels prepared 350-W microwave drying via varied sol–gel processes with different solvent volume (V).

V/mL	*S*_p_^a^ (m^2^/g)	*V*_pore_^b^ (cc/g)	*S*_micro_^c^ (m^2^/g)	*V*_micro_^c^ (cc/g)
1	52	0.06	28	0.01
2	784	1.3	584	0.3
3	784	1.9	596	0.3
4	611	3.3	455	0.2
5	714	3.0	515	0.3

^a^ Brunauer–Emmett–Teller specific surface area. ^b^ Pore volume calculated by NLDFT method [34] from the adsorption branch. ^c^ Micropore surface and volume calculated by the Dubinin–Astakhov method [35].

**Table 5 polymers-11-00375-t005:** Pore structure data of MSQ aerogels prepared by varied drying methods.

Drying Method	Drying Time	*S*_p_^a^ (m^2^/g)	*V*_pore_^b^ (cc/g)	*S*_micro_^c^ (m^2^/g)	*V*_micro_^c^ (cc/g)
700-W microwave drying	32 min	796	2.1	598	0.3
500-W microwave drying	33 min	821	2.2	617	0.3
350-W microwave drying	36 min	784	1.9	596	0.3
200-W microwave drying	55 min	783	2.1	599	0.2
40 °C oven drying	24 h	714	2.8	491	0.3
indoor evaporation	≥48h	795	2.4	610	0.3

^a^ Brunauer–Emmett–Teller specific surface area. ^b^ Pore volume calculated by NLDFT method [34] from the adsorption branch. ^c^ Micropore surface and volume calculated by the Dubinin–Astakhov method [35].

**Table 6 polymers-11-00375-t006:** Pore structure data of MSQ aerogels at varied heat treatment temperatures.

Temperature (°C)	*S*_p_^a^ (m^2^/g)	*V*_pore_^b^ (cc/g)	*S*_micro_^c^ (m^2^/g)	*V*_micro_^c^ (cc/g)
Without heat treatment	821	2.2	617	0.3
200	788	2.1	599	0.3
300	738	2.1	545	0.3
400	580	1.5	432	0.2
600	471	1.1	291	0.2
800	322	0.9	210	0.1

^a^ Brunauer–Emmett–Teller specific surface area. ^b^ Pore volume calculated by NLDFT method [34] from the adsorption branch. ^c^ Micropore surface and volume calculated by the Dubinin–Astakhov method [35].

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
