# Peer review of "Facile Synthesis of Methylsilsesquioxane Aerogels with Uniform Mesopores by Microwave Drying"

_polymers, 2019, doi:10.3390/polym11020375_

Round 1

Reviewer 1 Report

This manuscript describes process optimization of parameters to form aerogels using microwave drying method. While I commend the authors on their investigation on the various process parameters affecting the morphology of the resulting aerogels, I think that this manuscript needs to be revised according to the suggestions below before acceptance. 

Pg 3, line 73. Figure 2f should be Figure 1f.

Figure 2f. This figure was presented in terms of hydrolysis and polymerization mechanism. What do you mean by velocity of hydrolysis? How do you define it? What is the unit and how do we measure it? Is it volume change? Please clarify. Or is Figure 2f is presented fully for illustrative purpose?

In Section 3.2 regarding the preparation of aerogels, please be more elaborate on the specific example of aerogel production. For example, as an example, how much are the amount of each of the individual components being added to form the gels. x g of MTM, x g of CTAC, etc. 

When discussing the amount of various variables throughout the manuscript, please describe in terms of volume percentage for example Figure 1. How would the readers know the volume of gelation agent relative to what? Another example, for Figure 7, authors used volume to describe the various preparation conditions, but what is the volume relative to? It is very unclear. In another example on the condition in Figure 4, what is the overall volume fraction or mass fraction of M/W mixture used. My suggestion to remedy this issue is to put out a table of composition of all materials synthesized and presented in the manuscript. It would also be useful to refer to each of these examples with a sample name. In this way, the readers can refer back to the table and know exactly what sample corresponds to which conditions. 

Also, in Figure 4, 5, 7, 9, 12, please be more specific on the condition of fabrication of each of these samples. What are the ratio of M/W used, the pH used, the microwave power used for drying and heat treatment involved. I think it is best that we have a table listing all the condition and you can give sample names and refer to their names in each of those Figures mentioned. I think this is clearly needed. 

Author Response

Manuscript ID polymers-434054

Title: Facile Synthesis of Methylsilsesquioxane Aerogels with Uniform Mesopores by Microwave Drying

Author(s): Xingzhong Guo * , Jiaqi Shan , Wei Lei , Ronghua Ding , Yun Zhang , Hui Yang

Response to Referee #1

Thank you very much for your constructive and valid comments.

Q1. Pg 3, line 73. Figure 2f should be Figure 1f.

A1: Thank you for the suggestion, it has been revised.

Q2. Figure 2f. This figure was presented in terms of hydrolysis and polymerization mechanism. What do you mean by velocity of hydrolysis? How do you define it? What is the unit and how do we measure it? Is it volume change? Please clarify. Or is Figure 2f is presented fully for illustrative purpose?

A2: Thank you for the suggestion. This figure is presented fully for illustrative purpose. We made this diagrammatic sketch to summarize effect of pH for hydrolysis and polymerization reaction of precursor mentioned in references 32 and 33 in order to show the relationship between hydrolysis and polymerization reaction during different pH conditions.

Q3. In Section 3.2 regarding the preparation of aerogels, please be more elaborate on the specific example of aerogel production. For example, as an example, how much are the amount of each of the individual components being added to form the gels. x g of MTM, x g of CTAC, etc.

A3: Thank you for the suggestion, it has been revised in our revised manuscript. (Page 2 line 81)

Q4. When discussing the amount of various variables throughout the manuscript, please describe in terms of volume percentage for example Figure 1. How would the readers know the volume of gelation agent relative to what? Another example, for Figure 7, authors used volume to describe the various preparation conditions, but what is the volume relative to? It is very unclear. In another example on the condition in Figure 4, what is the overall volume fraction or mass fraction of M/W mixture used. My suggestion to remedy this issue is to put out a table of composition of all materials synthesized and presented in the manuscript. It would also be useful to refer to each of these examples with a sample name. In this way, the readers can refer back to the table and know exactly what sample corresponds to which conditions.

A4: Thank you for the suggestion, we have made a table of composition of all samples presented in the manuscript. (Page 2 line 81 and Page 3 line 93)

Q5. Also, in Figure 4, 5, 7, 9, 12, please be more specific on the condition of fabrication of each of these samples. What are the ratio of M/W used, the pH used, the microwave power used for drying and heat treatment involved. I think it is best that we have a table listing all the condition and you can give sample names and refer to their names in each of those Figures mentioned. I think this is clearly needed.

A5: Thank you for the suggestion, we have made a table of composition of all samples presented in the manuscript. (Page 2 line 81 and Page 3 line 93)

Author Response

Manuscript ID polymers-434054

Title: Facile Synthesis of Methylsilsesquioxane Aerogels with Uniform Mesopores by Microwave Drying

Author(s): Xingzhong Guo * , Jiaqi Shan , Wei Lei , Ronghua Ding , Yun Zhang , Hui Yang

Response to Referee #2

Thank you very much for your constructive and valid comments.

Q1. The authors describe in detail, the effect of pH, M/W, microwave power etc., on the pore size of MSQ aerogels. It would significantly enhance the impact of the paper if the authors can make a coup de grâce plot summarizing the ideal conditions needed to make MSQ aerogel with desirable pore size and surface area.

A1: Thank you for the suggestion. As we know, MSQ aerogel can be applied in various fields. In different fields, desirable pore size and surface area of MSQ aerogel can be different. In our report, we can only summarize an ideal condition making MSQ aerogel with desirable pore size and surface area from the point of view of micro-morphology and specific surface. When the mole ratios of MTMS : HCl : H2O : MeOH : CTAC : PO is 1 : 7.13*10-4 : 3.96 : 1.76 : 0.035 : 0.68 and the microwave power is 500 W, MSQ aerogel with a specific surface area of 821 m2/g and a mesopore size of 20 nm can be obtained. (Page 13 line 320)

Q2. There is literature suggesting the use of microwave drying for the preparation of aerogels. I did not find enough citations in this article suggesting the same. I would recommend the authors to find relevant literature and add appropriate references. Below is one such example I found.

a. Silica based aerogel-like materials obtained by quick microwave drying L. Duraes, T. Matias, R. Patricio, A. Portugal, DOI 10.1002/mawe.201300140

A2: Thank you for the suggestion, it is our negligence that we did not cite enough references about microwave drying. In fact, before we did the research, we had two references about microwave drying of aerogels. One is the article you have recommended, and another is Hygro-thermal properties of silica aerogel blankets dried using microwave heating for building thermal insulation Kevin insulation Kevin Nocentinia, Patrick Achard, Pascal Biwolea, Marina Stipeticca, DOI 10.1016/j.enbuild.2017.10.024. According to your suggestion, we have added these two articles to the references (Page 2, line 57, reference 29 and 30).

Q3. The organization of the paper is not conducive to follow the flow of ideas. I would recommend making Experimental Section as section 2, followed by Results and Discussion as section 3. This will allow the reader to become familiar with the materials used and experimental techniques carried out before understanding results.

a. Also, section 2.3 Discussion on microwave drying mechanism of MSQ aerogel, should be part of experimental section rather than in Results and Discussion section.

A3: Thank you for the suggestion, we accept your suggestion to make experimental section as section 2 (it has been revised in the Manuscript). However, we think section 2.3 Discussion on microwave drying mechanism of MSQ aerogel should be part of result and discussion after our careful consideration, because discussion on microwave drying mechanism also is a prospect for follow-up research on microwave drying.

Q4. Authors mention (in the legend below all the tables) the use of NLDFT to determine pore volume and Dubinin-Astakhov method to determine micropore surface and volume. It would be useful for the readers if the authors can describe these methods in Experimental section with appropriate references.

A4: Thank you for the suggestion. NLDFT is universally recognized as an N2 adsorption model to calculate pore volume of porous materials which is embedded in test software of BET. We get the data of pore volume of MSQ aerogel from the test report of a fully automatic specific surface and porosity analyzer. And Dubinin-Astakhov method is a model to calculate micropore surface and volume of porous materials. According to your suggestion, we have added two articles about NLDFT and Dubinin-Astakhov model into references (Page 15, line 418 and 420, reference 34 and 35).

Q5. (Minor correction) In Figure 11, the DTA/TG curve and IR Spectrum figures are not clearly visible. I would recommend the authors to add a clear image before final publication.

A5: Thank you for the suggestion, it has been revised in the manuscript. (Page 11, line 252)

Round 2

Reviewer 1 Report

Comments have been addressed satisfactorily and the manuscript can now be accepted for publication. 

Reviewer 2 Report

The authors have significantly improved the manuscript by accepting reviewersr suggestion. I recommend to agree the manuscript in the present form.